# Research Progress of Light Wavelength Conversion Materials and Their Applications in Functional Agricultural Films

**DOI:** 10.3390/polym14050851

**Published:** 2022-02-22

**Authors:** Yi Liu, Zhiguo Gui, Jialei Liu

**Affiliations:** 1School of Information and Communication Engineering, North University of China, Taiyuan 030051, China; liuyi@nuc.edu.cn (Y.L.); gzgtg@163.com (Z.G.); 2Institute of Environment and Sustainable Development in Agriculture, CAAS, Beijing 100081, China

**Keywords:** light conversion, agricultural film, luminescent materials, agricultural optics, optical modulation

## Abstract

As new fluorescent materials, light wavelength conversion materials (light conversion agents) have attracted increasing attention from scientific researchers and agricultural materials companies due to their potential advantages in efficiently utilizing solar energy and increasing crop yield. According to the material properties, the light conversion agents can be divided into fluorescent dyes, organic rare-earth complexes, and inorganic rare-earth complexes. The current researches indicates that the fluorescent dyes have relatively high production costs, poor light stability, difficult degradation processes, and easily cause pollution to the ecological environment. The organic rare-earth complexes have short luminescence times, high production costs, and suffer from rapid decreases in luminescence intensity. Compared with fluorescent dyes and organic rare-earth complexes, although rare-earth inorganic complexes have high luminous efficiency, stable chemical properties, and better spectral matching performance, the existing inorganic light conversion agents have relatively poor dispersibility in agricultural films. According to the research on light conversion agents at home and abroad in recent years, this paper first introduces the three common light conversion agents, namely fluorescent dyes, organic rare-earth complexes, and inorganic rare-earth complexes, as well as their uses in agricultural films and their mechanisms of light conversion. At the same time, the preparation methods, advantages, disadvantages, and existing problems of various light conversion agents are classified and explained. Finally, we predict the development trends for light conversion agents in the future by considering six aspects, namely efficiency, cost, compatibility with greenhouse films, light matching, and light transmittance, in order to provide a reference for the preparation of stable and efficient light conversion agent materials.

## 1. Introduction

Agricultural films are polymer materials, such as polyethylene (PE), polyvinyl chloride (PVC), Poly-(butylene adipate-co-terephthalate) (PBAT), Ethylene-vinyl acetate copolymer (EVA), etc. Pure polymer materials have many limitations in practical application, so improving the properties of polymer materials has always been the direction of interest for researchers or agricultural materials companies. Three common light wavelength conversion materials (light conversion agents), namely fluorescent dyes, organic rare-earth complexes, and inorganic rare-earth complexes, have potential advantages in potential advantages in efficiently utilizing solar energy and increasing crop yield, so their combination with agricultural film is an inevitable development trend. However, whereas there are still many deficiencies in six aspects: efficiency, cost, compatibility with greenhouse films, light matching, and light transmittance, and this greatly limits the application of multifunctional agricultural film. Therefore, the research on agricultural film with polymer as carrier and its ideal added components is a cause worthy of full-time investment of scientific researchers.

Else, Functionalization has always been an important research direction of polymer materials. The organic combination of light conversion agent and polymer materials realizes the effective regulation of polymer materials on the solar spectrum, which is an important means to realize the optical functionalization of normal polymer materials like PE, PP, PVC, PBAT and so on.

Solar energy is an important energy source and is a necessary condition for plant photosynthesis. The scientific literature shows that the blue-violet (400–480 nm) and red-orange light (600–700 nm) in sunlight can promote the growth and development of crops. The blue-violet light is beneficial in increasing the protein content of crops, while the red-orange light increases the synthesis of crop carbohydrates [1,2,3,4,5,6,7,8,9,10]. However, the emission of the ultraviolet light (280–390 nm) is not only unfavorable to the growth of plants, but also increases plant diseases and insect pests. Some sunlight will be reflected when it is emitted to the surface of the earth due to the presence of the atmosphere, which leads to this part of the sunlight being wasted. In recent years, the research in the field of agricultural film materials has mainly focused on the development of agricultural films with high-efficiency light conversion performance in order to make greater use of sunlight. The high-efficiency light conversion performance of agricultural film materials is mainly achieved using light conversion agents. A light conversion agent is a fluorescent material that can convert UV light into blue-violet light or red-orange light. In recent years, it has attracted extensive attention and has been actively developed. An agricultural film added with a light conversion agent is called a light conversion film, which can convert the lower active wavelength band in photosynthesis to the higher active wavelength band (400–700 nm), in turn changing the ultraviolet (280–380 nm) and yellow green (510–580 nm) wavelengths of sunlight into the blue-violet (400–480 nm) and red-orange (600–700 nm) wavelengths required for plant photosynthesis, improving crop yield and quality [11,12,13,14,15,16,17,18,19,20]. Therefore, based on the research on light conversion agents at home and abroad in recent years, this paper classifies the types of light conversion agents and their preparation methods, summarizes the applications for light conversion films in plants, and predicts the future development directions for light conversion agents and light conversion films.

## 2. Classification of Light Conversion Agents

Light conversion agents can be divided into green-to-red agents, ultraviolet-to-red agents, and ultraviolet-to-blue agents according to their light conversion properties. Regarding the luminescent properties, these agents can be categorized into red luminescent agents, blue luminescent agents, and red-blue compound agents. Regarding the material properties, these can be divided into fluorescent dyes, organic rare-earth complexes, and inorganic rare-earth complexes.

### 2.1. Fluorescent Dyes

Fluorescent dyes are compound dyes with good fluorescence emission performance. They can absorb light of certain wavelengths and then emit other substances with longer wavelengths than the absorbed light wavelength. Most of the fluorescent dyes contain benzene rings or heterocycles and have conjugated double bonds. Due to the good compatibility between organic fluorescent dyes and substrates, the light conversion films made of organic fluorescent dyes have the advantages of uniform distribution and convenient operation.

Yu et al. [21] prepared three kinds of UV-to-blue light conversion agents, namely 2,4,6-tris (diphenylamino) 1,3,5-triazine (TRZ), 4,6-tris (di-p-tolylamine) 1,3,5-triazine (TRZME), and 2,4,6-tris (4-methoxy-3-methyl-N-phenylaniline) 1,3,5-triazine (TRAOME), and made light conversion films by doping these light conversion agents into the polyvinyl chloride. The synthetic route of light conversion agents (S-triazine compounds) is shown in Figure 1. The experimental results showed that compared with the PVC film without a light conversion agent, the light conversion films had lower transmittance to UV light (190–335 nm, as shown in Table 1) and could change the absorbed ultraviolet light into blue light with a fluorescence emission peak at 400–460 nm. At the same time, this did not cause a reduction in visible light transmittance.

Kyung et al. [22] synthesized four 3- and 4-position naphthalimide blue fluorescent dyes with diacetylene bonds (Figure 2 shows the synthesis route) and coated them on the PE film. They studied the impacts of the diacetylene bond on the fluorescence properties of naphthalimide-based dyes and the application of the dyes in light conversion films. The study results showed that the four synthetic fluorescent dyes effectively enhanced the photosynthesis of blue light. The transmittance of the PE film coated with fluorescent dye 1 at 400–500 nm was 46.8% higher than that of the transparent film, which was in line with the result showing that the photosynthetic photon flux density showed an increase of 90% in transmittance in the blue region (400–500 nm). At the same time, it was found that by using the PE film coated with fluorescent dye 1, the lettuce showed significant increases in fresh weight, dry weight, and leaf area after three weeks of culture.

The spectrum of sunlight reaching the earth’s surface is between 290 and 3000 nm. Different components of light have different effects on plant growth. Blue-violet light in the wavelength of 400~480 nm can be strongly absorbed by chlorophyll and carotenoids to promote the growth of plant stems and leaves. Red-orange light in the wavelength of 600~700 nm is absorbed by chlorophyll, which can promote the fruit growth of plants. Yellow-green light with wavelengths of 500~600 nm has little contribution to photosynthesis. Near-ultraviolet light in the wavelength of 315~400 nm makes the plants shorter and the leaves thicker. Ultraviolet rays in the wavelength of 290~315 nm are harmful to most plants and can also lead to the aging of PE films. During photosynthesis, plants absorb sunlight through chlorophyll, decompose water, and release hydrogen (H^+^ and e^−^), reducing the absorbed carbon dioxide to organic matter. In this process, carbon dioxide and water are transformed into organic matter and solar energy is transformed into chemical energy, which is stored in organic matter. The light conversion films can accelerate this process in plants.

Wang et al. [23] designed and synthesized six triarylacrylonitrile compounds and one diarylacrylonitrile compound and prepared light conversion films doped with these compounds. In the simulated greenhouse environment, the luminescence agents had excellent light stability. However, after 1 month of outdoor radiation, their fluorescence intensity dropped to 17–40% of the basic intensity due to the closed-loop oxidation reaction (in summer). By comprehensively considering the above-mentioned photophysical and mechanical properties of the doped films, 2-([1,1′-biphenyl]-4-yl)-3,3-diphenylacrylonitrile can potentially be used as a light conversion agent for agricultural films in winter. Xiao et al. [24] used acetonitrile, absolute ethanol, and p-methoxybenzaldehyde as starting materials to synthesize 2,4,6-tris (p-methoxystyryl)-1,3,5-triazine (TMST) and studied the light conversion properties of TMST. The fluorescence spectrum of the TMST demonstrated that this compound emitted strong blue-violet light in the range of 410–480 nm, with the strongest emittance observed at 445 nm. As the blue light (410–480 nm) falls exactly within the blue-violet light absorption band of chlorophyll, this compound is expected to be used as a new blue light conversion agent. If it is applied to the agricultural film, it could effectively improve the utilization rate of light energy.

Previous studies have shown that fluorescent dye light conversion agents possess good light conversion efficiency. However, fluorescent-dye-type light conversion agents cannot accept long-term light irradiation because they are prone to chemical reactions, which may reduce the use time. They also involve high costs and difficult degradation processes and easily cause pollution to the natural environment, making them unfavorable for the development of ecological agriculture. Therefore, they are often used in combination with other light conversion agents in practical applications.

### 2.2. Organic Rare-Earth Complexes

Organic rare-earth complexes are light conversion agents formed by complexing the rare-earth metal ion as the luminescence center with the organic ligands [25]. Organic rare-earth complexes have great advantages, including high fluorescence intensity, easy dispersion, good light and color uniformity, and excellent photo-thermal stability [26].

Qiao et al. [27] took kaolinite as the main body and prepared luminescent clay (labeled with K-NMI-Eu and K-NMI-Tb, respectively) by inserting N-methylimidazole (NMI) molecules and Eu^3+^/Tb^3+^ salts into kaolinite. The research results showed that under the ultraviolet light, K-NMI-Eu had excellent ability to convert ultraviolet light to red light (as shown in Figure 3).

Wang et al. [28] synthesized Eu(TTA)_3_(TPPO)_2_ rare-earth europium compounds with the solution method, which absorbed ultraviolet light and then converted it into red light, then compounded this with PLA/PBAT to prepare a biodegradable light conversion agricultural film and studied its optical and mechanical properties. The results showed that when 0.1% EuTT was added, the PLA/PBAT/EuTT composite film showed better mechanical properties (see Table 2), with tensile strength values of 36.7 MPa and 25.2 MPa. The red fluorescence (617 nm) could be emitted under ultraviolet light (365 nm). Yu et al. [29] combined the rare-earth europium with different organic ligands to prepare two rare-earth light conversion agents Eu(DBM)_4_CPC and Eu(TTA)_3_(TPPO)_2_, and combined them with polylactic acid and polyadipate to prepare two different light conversion films. The results illustrated that the two light conversion films had excellent light conversion ability and were able to emit red fluorescence (617 nm) under ultraviolet light (365 nm). At the same time, the mechanical properties of the light conversion film were also improved, reaching 595.0%/460.9% in the longitudinal (MD) and transverse (TD), respectively (shown in Table 3). This was mainly because the addition of rare-earth complexes improved the fluidity of the melt and reduced the melt viscosity of the blend, which greatly increased the elongation at break of the films.

Zhao et al. [30] synthesized a series of thiophene trifluoropyruvate (HTTA), terephthalic acid (TPA), and Phen (III) complexes with the thermal reflux method. Through the elemental, infrared spectroscopy, scanning electron microscopy, and thermal stability analyses, the new Eu(TPA)(TTA)Phen and Eu_2_(TPA)(TTA)_4_Phen_2_ complexes were characterized. The results showed that the thermal stability of Eu (III) complexes increased in the following order: the mononuclear complex Eu(TTA)_3_Phen, binuclear complex Eu_2_(TPA)(TTA)_4_Phen_2_, and chain polynuclear complex Eu(TPA)(TTA). The formation of the binuclear–multinuclear structure of the new complex seems to be the reason for its enhanced thermal and optical stability (Figure 4 shows the chemical structural formula). In addition, corresponding to their formation, the fluorescence excitation spectra of these new complexes gave a wider excitation band than Eu(TTA)_3_Phen. Through the addition of Gd^3+^, increased in Eu^3+^ fluorescence in the new complexes was observed. When the addition of EuGd(TPA)(TTA)_4_Phen_2_ exceeded 0.5% (mass fraction), bright red luminescent plastics could be obtained. Xi et al. [31] synthesized a new type of light conversion agent suitable for PE films, and characterized the morphology, particle size, dispersion performance, and compatibility of the new light conversion agent via transmission electron microscopy and scanning electron microscopy. The results indicated that with an average particle size range of 60–100 nm, the new light conversion agent was well dispersed in the PE film. In addition, the fluorescence spectrum confirmed that the new light conversion agent provided high-efficiency luminescence performance.

The organic rare-earth complexes are featured by short luminescence time, high cost, and fast luminescence intensity decline, while the characteristic luminescence obtained by organic complexes is not a good for with the absorption spectrum of plant chlorophyll, which requires more in-depth research. The structural characteristics of the high coordination of the rare-earth ions make it easy for them to form ion clusters during the coordination synthesis at high concentrations, resulting in concentration quenching and making it difficult to obtain luminescent complexes with higher intensity. In addition, it is hard to uniformly dope rare-earth compound particles into the polymer matrix during the preparation of organic rare-earth complexes, which usually results in the polymer having poor mechanical strength. Therefore, work in this field has focused on the development of various rare-earth complexes. Studies have shown that the organic ligands are able to promote the compatibility between rare-earth elements and polymers [32,33].

### 2.3. Inorganic Rare-Earth Complexes

Inorganic rare-earth complexes are light conversion agents prepared by doping inorganic substances with rare-earth ions, which have attracted widespread attention due to their low price and high temperature resistance [26].

Wu et al. [26] synthesized Sr_2_Si_5_N_8_:2% Eu phosphor via solid-phase reaction and prepared a light conversion film with Sr_2_Si_5_N_8_:2% Eu as the light conversion agent. The transmission electron microscopy results indicated that the average diameter of the light conversion agent was 500 nm and that the film structure was denser after the light conversion agent was added. The fluorescence spectra confirmed that the prepared light conversion film converted the blue-violet light into red light. Wang et al. [1] made CaCO_3_:Eu^3+^ phosphors modified with surfactants using the carbonization method and prepared a PE film with the modified CaCO_3_:Eu^3+^ phosphor. The experimental results illustrated that compared with the oleic acid, stearyl alcohol phosphoric acid, and sodium lauryl sulfate, sodium oleate was the best modifier for CaCO_3_:Eu^3+^ phosphor, as the CaCO_3_:Eu^3+^ phosphor modified by sodium oleate was evenly dispersed in the PE film. The modified PE film emitted uniform red light under the excitation of ultraviolet rays, while the modified PE film absorbed more ultraviolet light and changed it into enhanced red light, which was beneficial to the plant growth. Zhu et al. [34] prepared SrAl_2_O_4_:Eu^2+^,Dy^3+^ particles through solid-phase reaction and combined the light conversion agent with SrAl_2_O_4_:Eu^2+^,Dy^3+^ through YsiX3 to prepare a luminescent material, namely the SrAl_2_O_4_:Eu^2+^,Dy^3+^ light conversion agent. The results showed that the light conversion agent was completely coated on the surface of the phosphor and that the coated light conversion agent was very dense and had no holes. Under ultraviolet excitation, the emission spectrum gave a broadband range of 450 nm to 650 nm, while the excitation intensity levels were highest at 520 nm (SrAl_2_O_4_:Eu^2+^,Dy^3+^) and 600 nm (SrAl_2_O_4_:Eu^2+^,Dy^3+^ light conversion agent). Additionally, the colors of the SrAl_2_O_4_:Eu^2+^,Dy^3+^ light conversion agent were mainly in the orange-red area. Sun et al. [19] prepared SrB_4_O_7_:Sm^2^ phosphor via high-temperature solid-phase reaction and studied its luminescence properties. The test results and theoretical calculations demonstrated that the SrB_4_O_7_:Sm^2^ phosphor showed a strong emission at 685 nm under excitation of 356 nm. The best doping concentration of SrB_4_O_7_:Sm^2+^ in Sm^2+^ phosphor was 0.05. The d-d interaction played a major role in the Sm^2+^ quenching mechanism in the SrB_4_O_7_:Sm^2+^ phosphor. It is worth mentioning that the decay times were almost unchanged at different concentrations. The above results show that SrB_4_O_7_:Sm^2+^ phosphor can be used as a new type of light conversion agent to improve the photosynthesis of plants. In order to study the effects of the light conversion agent on the luminescence properties of rare-earth luminescent fibers, Zhu et al. [35] used rare-earth strontium aluminate as the luminescent material and fiber-forming polymers (e.g., polyethylene terephthalate (PET)) as the substrate to prepare several rare-earth strontium aluminate luminescent fibers containing different light conversion agents. The scanning electron microscopy (SEM) and X-ray diffraction (XRD) results showed that there was no significant difference between the surfaces of the luminescent fibers without the light conversion agent and the surfaces of the luminescent fibers containing the light conversion agent. The SrAl_2_O_4_:Eu^2+^,Dy^3+^ phase in the fibers was not destroyed by the complicated manufacturing process and the luminescent fibers that were formed were composed of irregular particles. Under ultraviolet excitation, the luminescent fibers displayed yellow-green and orange-red emission bands, and the maximum wavelengths at 520 nm and 600 nm were derived from SrAl_2_O_4_:Eu^2+^,Dy^3+^ light conversion agent. In addition, the emission colors of the luminescent fibers were easily adjusted from yellow-green to orange-red by adding different light conversion agents. Therefore, this material has potential application prospects in many fields.

Compared with organic compounds and fluorescent dyes, inorganic light conversion agents have been widely used due to their low price, easy preparation and storage, oxidation resistance, and high temperature resistance. These types of light energy conversion materials overcome the poor light stability of fluorescent dyes and high production costs of rare-earth organic complexes. In addition, they have high luminous efficiency and good stability, a wide light wavelength conversion range, and an emission spectrum that can match the absorption spectrum of plant chlorophyll. However, the biggest disadvantages of existing inorganic light conversion agents are poor dispersion and easy aggregation in agricultural plastic films, which makes the emission of light conversion films inhomogeneous and means they are not conducive to crop growth [36,37]. The reasons for the poor dispersion are that inorganic light conversion agents usually have polar, hydrophilic, and high free energy surfaces and are incompatible with certain non-polar, hydrophobic, and low free energy polymer matrices.

## 3. Preparation of Light Conversion Agents

Currently, light conversion agents are mainly prepared via the high-temperature solid-phase synthesis method, the sol–gel method, the hydrothermal synthesis method, the co-precipitation method, the combustion method, the spray pyrolysis method, and the microwave radiation method. This article mainly discusses the first three methods.

### 3.1. High-Temperature Solid-Phase Synthesis Method

The high-temperature solid-phase method is used to grind the raw materials in a certain proportion and then calcine them at a certain temperature for a certain time in a certain atmosphere. Afterwards, the materials can be cooled, ground, and sieved to obtain the final required light conversion material.

Zhang et al. [38] synthesized a series of Ca_2_Sr(PO4)_2_:Ce^3+^, Mn^2+^, and Na^+^ phosphors through high-temperature solid-phase reactions and studied their luminescence properties and water resistance. The PLE, PL, and DR spectra and the attenuation curves indicated strong energy transfer from Ce^3+^ emitted by near-ultraviolet light to Mn^2+^ emitted by red light. The diffuse reflectance and UV–visible absorption spectra showed (Figure 5) that the Ca_2_Sr(PO4)_2_:Ce^3+^, Mn^2+^, and Na^+^ phosphors had strong absorption in the near-ultraviolet range of 320 nm. The absorbance increased with the increase in Mn^2+^ content from 0.10 to 0.15, which was beneficial to the anti-aging ability of the agricultural films. Additionally, the absolute QE value of the co-doped phosphor reached 94%, representing good stability in water. Therefore, the Ca_2_Sr(PO4)_2_:Ce^3+^, Mn^2+^, and Na^+^ phosphors have potential application prospects as light conversion materials in agricultural films. Mondal et al. [39] used zinc oxide, molybdenum oxide, erbium oxide, yttrium oxide, and lithium carbonate as raw materials to synthesize a series of Er^3+^/Yb^3+^/Li^+^/K^+^:ZnMoO_4_ phosphors via solid-phase synthesis, then characterized and analyzed their structures and optical properties. The experiments demonstrated that under excitation at 980 nm, the Er^3+^-Yb^3+^-doped phosphor showed strong green light emission at 531 and 553 nm, corresponding to the ^2^H_11/2_/^4^S_3/2_→^4^I_15/2_ transition. It also displayed weak blue and red emissions at 408, 474, and 656 nm, corresponding to the ^2^H_9/2_→^4^I_15/2_, ^4^F_7/2_→^4^I_15/2_, and ^4^F_9/2_→^4^I_15/2_ transitions, respectively. This study also showed that the mixing of Er^3+^-Yb^3+^-based phosphors and K^+^ ions improved the temperature sensitivity and temperature range. He et al. [40] first prepared carbon dots (CD) via high-temperature solid-phase reaction and then directly recombined CD and Eu^3+^ solutions instead of using the traditional method based on Eu^3+^ coordination compounds. They then successfully adjusted the photoluminescence characteristics by controlling the ratio of CD to Eu^3+^, which met the variable light demands of the different plant species. Finally, they inserted Eu^3+^ ions and carbon dots (CDs) into a polyvinyl alcohol (PVA) solution to make blue and red light conversion films (Eu^3+^/CD/PVA films), which had excellent visible light transmittance and ultraviolet absorption characteristics. It is worth mentioning that the films showed broad application prospects because of their good stability and reusability.

### 3.2. Sol–Gel Method

The sol–gel method is a soft chemistry method that mixes compounds containing highly chemically active components in a solution uniformly to form a precursor, and then forms a transparent and stable sol system based on the hydrolysis and condensation reaction. The sol colloidal particles are slowly polymerized to form a gel with a three-dimensional network structure. The solvent loses its fluidity and fills in the network structure of the gel. Finally, the required material is prepared through drying and high-temperature sintering.

Liu et al. [41] first used the sol–gel method to make a series of Sr_2_MgSi_2_O_7_ particles and then prepared SOG/SMS composite films containing 5 wt% SMS particles with the one-step photolithography method (see Figure 6 for the preparation process). Finally, they characterized and analyzed their performance using a scanning electron microscope, fluorescence microscope, and fluorescence spectrophotometer. The results showed that the composite film had good adhesion and emitted blue fluorescence due to the characteristic transition from the 4f^6^5d excited state to the 4f^7^ ground state of the Eu^2+^ ion in the main body of Sr_2_MgSi_2_O_7_. Xue et al. [42] successfully prepared SiO_2_-coated Sr_4_Al_14_O_25_:Eu^2+^,Dy^3+^–LCA composite yellow phosphor using the sol–gel method. The light conversion agent and SiO_2_ were combined with strontium aluminate through Al-O-Si bonds and the silane coupling agent, respectively. The composite yellow phosphor was able to effectively absorb light in the range of 280 to 570 nm, and the emission peak intensity at 586 nm was much higher than the emission peak intensity at 493 nm. Therefore, the composite phosphor appeared yellow in the dark. Additionally, the light conversion agent was not only bonded with strontium aluminate but also interacted with SiO_2_ through hydrogen bonding and electrostatic interaction, causing the LCA chromophore to be positively charged. This interaction increased the excitation of the light conversion agent and the emission peak changed from 606 nm (orange-red) to 596 nm (yellow). This reflected the important role of the SiO_2_ coating in the emission of composite yellow phosphors. Moreover, due to being coated with silicon dioxide, the SiO_2_-Sr_4_A1_14_O_25_:Eu^2+^,Dy^3+^/LCA composite yellow phosphors had good water resistance. In order to obtain high-efficiency phosphors that emitted long wavelengths, Zhang et al. [38] successfully prepared SiO_2_-Sr_4_A1_14_O_25_:Eu^2+^,Dy^3+^/LCA (light conversion agent) with the sol–gel method. The migration phenomenon was analyzed via fluorescence and FTIR spectra and the blue shift was caused by the chemical bond between LCA and SiO_2_. The study also found that when the SiO_2_ content was changed within the range of 0–2 wt%, there was a significant blue shift; when the silica content was 6 wt%, the quantum yield reached the maximum. Since the silica coated on the composite phosphor through chemical bonding and the silica layer affected the SiO_2_-Sr_4_A1_14_O_25_:Eu^2+^,Dy^3+^ and LCA, the color of SiO_2_-Sr_4_A1_14_O_25_:Eu^2+^,Dy^3+^/LCA phosphor was altered in the yellow-red range with the change in silica content. This study indicated that SiO_2_-Sr_4_A1_14_O_25_:Eu^2+^,Dy^3+^/LCA phosphor could be used as a suitable luminescent material.

### 3.3. Hydrothermal Synthesis

The hydrothermal synthesis method is an effective method that uses water as a solvent in a special closed reactor (autoclave) to heat and pressurize the reaction system (or autogenous vapor pressure) in order to achieve a high-temperature and high-pressure environment in a closed state to dissolve and recrystallize insoluble substances for inorganic synthesis and material processing.

Mao [43] prepared two red-emitting MOFs with rare-earth metal europium as the metal luminescence center and with terephthalic acid or trimellitic acid as the organic–ligand chain using the hydrothermal method. The shift of the characteristic peaks of the functional group shown via FT-IR illustrated the coordination reaction between the organic ligands and the metal center. The micromorphology of the MOF showed that the MOF structure of terephthalic acid was stacked with many different blocks, while the pyromellitic acid showed a rod structure. Additionally, the change in the fluorescence peak intensity was used to reveal the impacts of the time and temperature of the hydrothermal synthesis reaction on the fluorescence intensity of the MOF, with an optimal reaction time of 48 h and optimal temperatures of 110 °C for terephthalic acid and 120 °C for trimellitic acid. At the same time, after thermogravimetric characterization, it was found that the thermal stability of the terephthalic acid MOF was relatively better than that of the trimellitic acid MOF. Most important of all, both MOFs showed good red light emission near 615 nm and were well dispersed in the organic film (as seen in Figure 7). This experiment provided two more affordable organic ligand chains and the prepared MOF structures showed excellent red light emission performance. Xu [26] prepared Re:KYW (Re = Eu^3+^, Sm^3+^, Eu^3+^/Sm^3+^) phosphors using the hydrothermal method to assist the organic active agents and studied the impacts of controllable factors on the luminescent and light conversion performance of KYW phosphors. The fluorescence results for Re:KYW (Re = Eu^3+^, Sm^3+^, Eu^3+^/Sm^3+^) explained that the morphology differences observed for KYW with different surfactants, which had an effect on the fluorescence properties. Among them, the sample with 10 mL acetylacetone had the strongest luminescence intensity. The excitation wavelength of Eu^3+^ ion was 397 nm and the doping concentration was 45%, which showed high-intensity red light emission (612 nm) and the best fluorescence performance. The results also showed that under certain conditions, the higher the crystallinity of the phosphor, the better the dispersion. This led to increases in the specific surface area of the phosphor and the integrity of the flower sphere, improving the fluorescence performance. Wu et al. [44] used a mixed solution including a CeO_2_-TiO_2_ precursor and ATO particles to deposit a Sb-doped SnO_2_(ATO)-(CeO_2_-TiO_2_) film on a glass substrate using a sol–gel dip coating process, then prepared ATO particles using the low-temperature hydrothermal synthesis method. The optical properties of the film were characterized by ultraviolet–visible light transmission and infrared reflectance spectroscopy. The sheet resistance of the ATO particles and the film was measured using a rubber sheet press and a four-point probe, while the surface morphology and structure of the film were analyzed using 3D Digitale Mikroskop and X-ray diffraction methods, respectively. It was found that the ATO precursor solution was completely weightless at about 500 °C, allowing ATO particles to be obtained, which confirmed that it had the same rutile lattice structure as SnO_2_. Moreover, the glass substrate coated with the ATO-(CeO_2_-TiO_2_) film showed better performance in shielding ultraviolet rays (almost 100%), visible light transmittance (70%), and infrared reflection (>30%).

## 4. Application of Light Wavelength Conversion Greenhouse Films

Recently, the application of light conversion films in greenhouse and field planting has gradually increased. Through the conversion of ultraviolet light into blue-violet light or red-orange light, the utilization of sunlight can be significantly improved, thereby providing an optical environment for crops. The increases in red and blue light promote the growth of crops and the accumulation of nutrients and play an important role in increasing crop yield and improving crop quality. In this section, the application effects of light conversion films are discussed in detail.

Many scholars have studied and improved the quality of light conversion films and such kind of light conversion films played a central role in greenhouse and field planting research. Light conversion films can convert ultraviolet light into blue-violet light and red-orange light, which are useful for crops, or can change yellow-green light into red-orange light, which changes the quality of the light passing through the film. This not only increases the leaf area, vine spread, plant height, and petiole length of plants, meaning the leaves have good ductility, but also effectively reduces the ecological pollution at the end of the season. In Gao’s study, a transplanted film used as a modified material in greenhouse lettuce cultivation had a beneficial effect in improving the light energy use efficiency for crop production [45]. The research by Yang et al. showed that a translucent film had great application prospects in field crop cultivation, as it promoted plant growth and increased the biological dry weight per plant [46]. Gao applied a light conversion film to seedling trays containing tobacco and tomato plants. The film was conducive to the growth of the seedlings, as they effectively increased the temperature in the membrane disk, improved the resistance and quality, and significantly promoted the aboveground biomass and root phenotypic parameters of the seedlings. Additionally, the time of seedling emergence was shortened by 2 d and the period of seedling establishment was 3 d ahead of schedule. The total root length, root surface area, and volume of tobacco seedlings at the seedling establishment stage were significantly increased by 44.88%, 69.87%, and 90.90%, respectively [47]. Zhang treated tomatoes with a rare-earth light conversion film, which significantly promoted the growth of green stems and leaves, increased the fruit yield, and improved the fruit quality. In addition, the fruits ripened 7d earlier and yield increased by 6.35% [48]. Song covered potato plants with light conversion films, which not only promoted potato emergence and maturity, but also increased the potato yield and output value by 8.91% [49]. Wen also covered tomato plants with a light conversion film, which significantly promoted the plant growth and development in the heliostat, with a 9.92% increase in tomato yield per unit area [50]. The sugar-to-acid ratio and contents of lycopene, soluble sugar, and VC in tomato fruit samples were also significantly increased, indicating that the addition of suitable light transfer agents to the trellis film could increase the proportions of red-orange and far-red light in the transmitted light. This promoted the growth and development of tomato plants and improved the yield and quality of fruit. In Liu’s studies [51], the light conversion film had a promotion effect on the growth of greenhouse strawberry plants. Firstly, the light conversion film caused a marked temperature increase. Compared with the normal film, the accumulated temperature in the greenhouse increased by 75.0 °C and the average daily temperature increased significantly by 0.39 °C. Secondly, the light conversion film promoted the growth and development of strawberry plants. The chlorophyll content and net photosynthetic rate of fruiting strawberry plants in the transplanted film sheds were markedly increased by 14.0% and 14.3%, respectively, and the use of the light conversion film meant strawberries could be picked about 10 d earlier. Thirdly, the light conversion film had improved the quality of the strawberries. The VC contents of strawberries grown under both transplanted films were significantly higher than when grown under normal film. Wen’s research found that the red light conversion film could significantly improve the yield and quality of cucumber fruit compared with the control. The red light conversion film promoted the growth of cucumber plants; prominently improved the plant height, stem diameter, internode length, and single fruit quality; and increased the yield by 16.78%. In addition, the contents of soluble sugar, free amino acids, and VC in cucumber fruit increased significantly [52]. Yan applied a broad-spectrum light conversion film to tomato plants, which enhanced the efficiency of the UV-to-red-light conversion and increased the total amount of red light emission. It also facilitated the photosynthesis of tomato plant leaves in the heliostat, promoted the growth of tomato seedlings and the absorption of nutrients and water, increased the yield, and improved fruit quality [53]. Sun’s research showed that the white light conversion film treatment significantly promoted the growth of tomato seedlings, increasing the strong seedling index by 67% and stem thickness by 44%, which was suitable for cultivation cover in South China and had good application prospects [54]. Li’s study proved that by increasing the ratio of red light to orange light under the red light conversion film, the high-temperature light effect had two advantages. On the one hand, it improved the photosynthetic efficiency of sweet pepper and promoted the absorption of nutrients and the synthesis of amino acids. On the other hand, it accelerated the degradation of chlorophyll and anthocyanin accumulation in fruit during discoloration by promoting carbon metabolism to improve the appearance and nutritional quality of the fruit. At the same time, the addition of suitable light conversion agents could increase the VC content of sweet pepper fruit [55].

## 5. Conclusions

Rare-earth inorganic complex light conversion agents have strong anti-UV aging functions, but their luminous efficiency is relatively weak, making them suitable for the preparation of long-life light conversion films and for use with crops with low demands for light intensity. Organic rare-earth complex light conversion agents have high light conversion efficiency, good matching with host polymers, and relatively high preparation costs. These agents will be important for the future development of light conversion agents and they are suitable for the preparation of high-performance light conversion films. Organic conjugated small-molecule light conversion agents have a wider absorption spectrum, poor light conversion efficiency, and poor aging resistance. They are more suitable for crops with special spectrum needs.

## 6. Future Development Trends for Light Conversion Agents

An analysis was performed to compare the three existing types of mature light conversion agents, namely inorganic agents, organic fluorescent dyes, and organometallic complexes, to identify their advantages and disadvantages. The current research on light conversion agents mainly focuses on the following six aspects. (1) Stability: The agricultural films prepared with the addition of light conversion agents should be able to be used under all weather conditions, meaning the light conversion agents need to have high stability, slow attenuation rates, and good weather resistance. (2) Light conversion efficiency: Light intensity levels and light times differ by regions, so the prepared light conversion agents should have higher light conversion efficiency. (3) Cost: The production costs of the light conversion agents need to be reduced, as the price of light conversion agents is one of the key indicators determining whether they will be adopted by companies and the common people. (4) Good compatibility with greenhouse films: Adding light conversion agents can filter ultraviolet light into red light required for plant photosynthesis and can increase greenhouse temperatures. Therefore, the prepared light conversion agents must be well dispersed in the agriculture films. (5) Good light matching with the growth of crops: The fluorescence spectrum emitted by the light conversion agent must be effectively matched with the absorption spectrum of plant photosynthesis to ensure that the red-light energy emitted by it can be used for plant photosynthesis. (6) High light transmittance: The addition of light conversion agents should improve the light conversion properties of agricultural films but not reduce their light transmittance. To solve the current problems, the focus should be on using the advantages of the existing light conversion agents and combining them with other materials with excellent fluorescence characteristics or fluorescent carriers to prepare new light conversion agents with strong red-light emission levels and better stability.

## Figures and Tables

**Figure 1 polymers-14-00851-f001:**
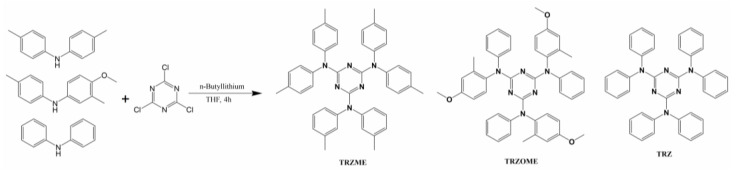
Synthesis route of S-triazine compounds.

**Figure 2 polymers-14-00851-f002:**
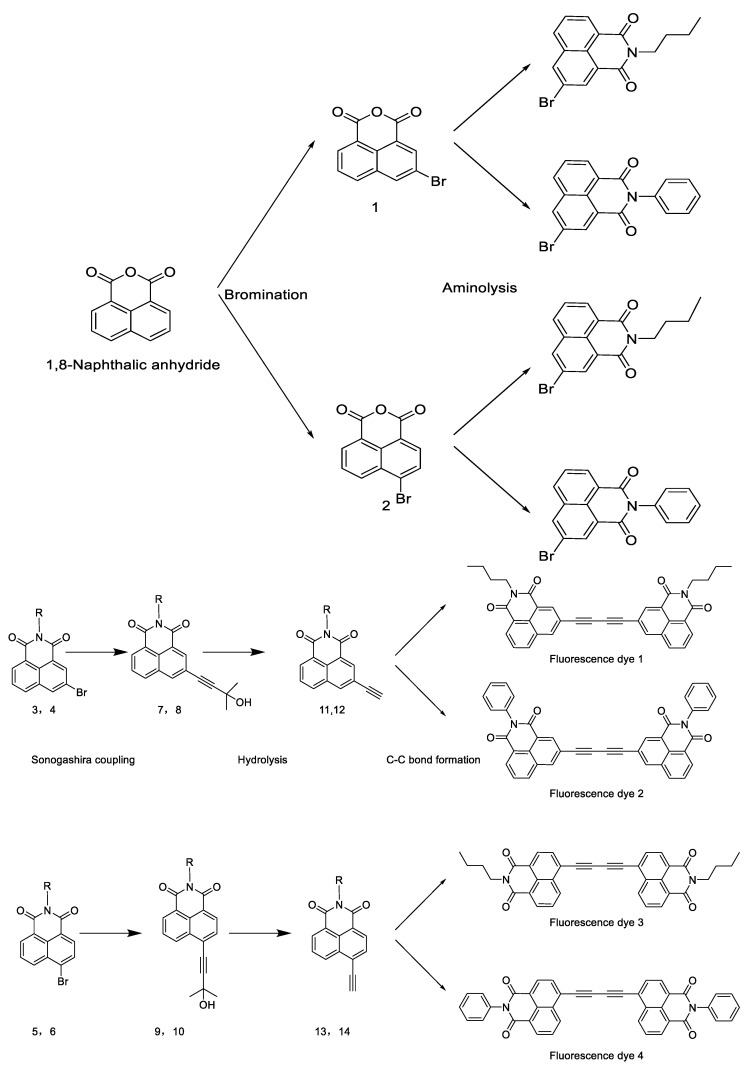
Synthesis routes of fluorescence dyes 1–4. Adapted with permission from ref. [22]. Copyright 2018 Science Direct.

**Figure 3 polymers-14-00851-f003:**
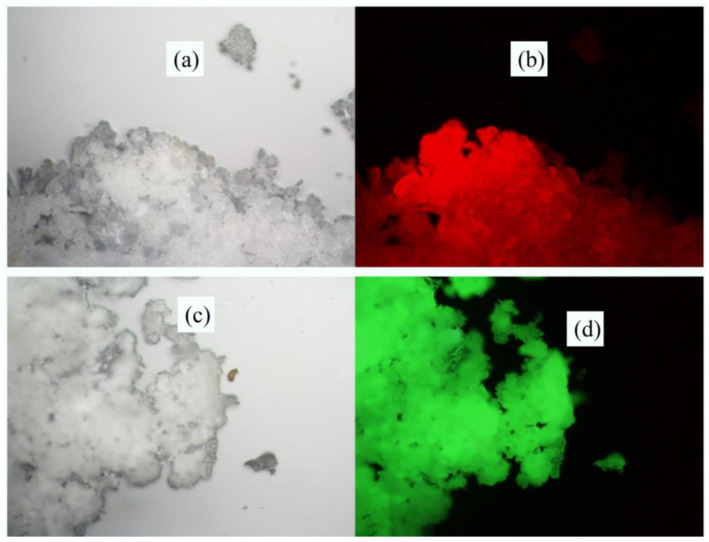
Photos of polycrystalline powders K-NMI-Eu and K-NMI-Tb exposed to ultraviolet light (λ = 330–380 Nm) under ambient lighting in a room (**a**,**c**) and at indoor temperature (**b**,**d**). Adapted with permission from ref. [27]. Copyright 2019 Science Direct.

**Figure 4 polymers-14-00851-f004:**
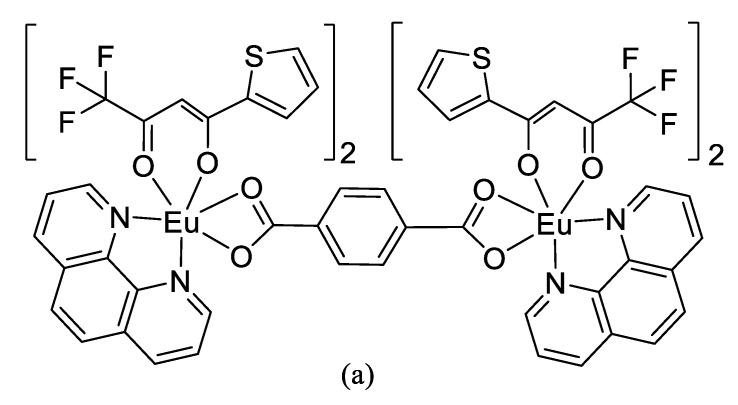
Chemical structures of complexes: (**a**) Eu_2_(TPA)(TTA)_4_Phen_2_; (**b**) Eu(TPA)(TTA)Phen. Adapted with permission from ref. [30]. Copyright 2007 Springlink.

**Figure 5 polymers-14-00851-f005:**
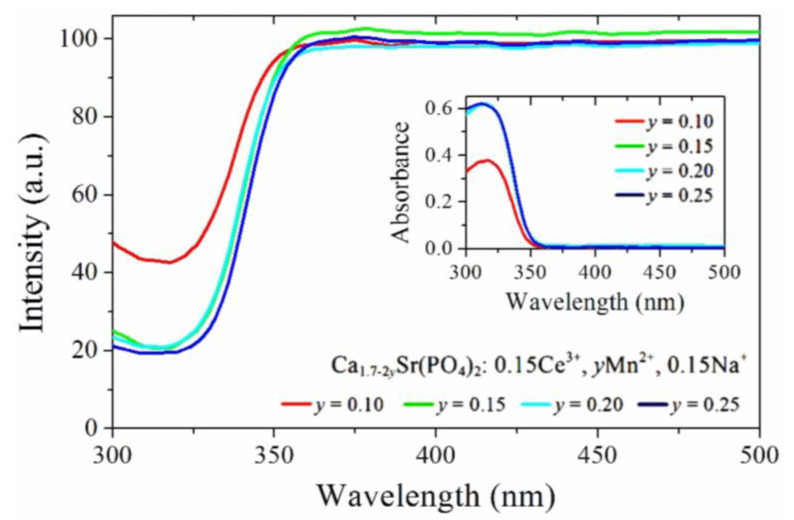
Diffuse reflectance and UV–Vis absorption spectra of Ca_1.7-y_Sr(PO_4_)_2_:0.15Ce^3+^, yMn^2+^, and 0.15Na^+^. Adapted with permission from ref. [38]. Copyright 2017 Royal Society of Chemistry.

**Figure 6 polymers-14-00851-f006:**
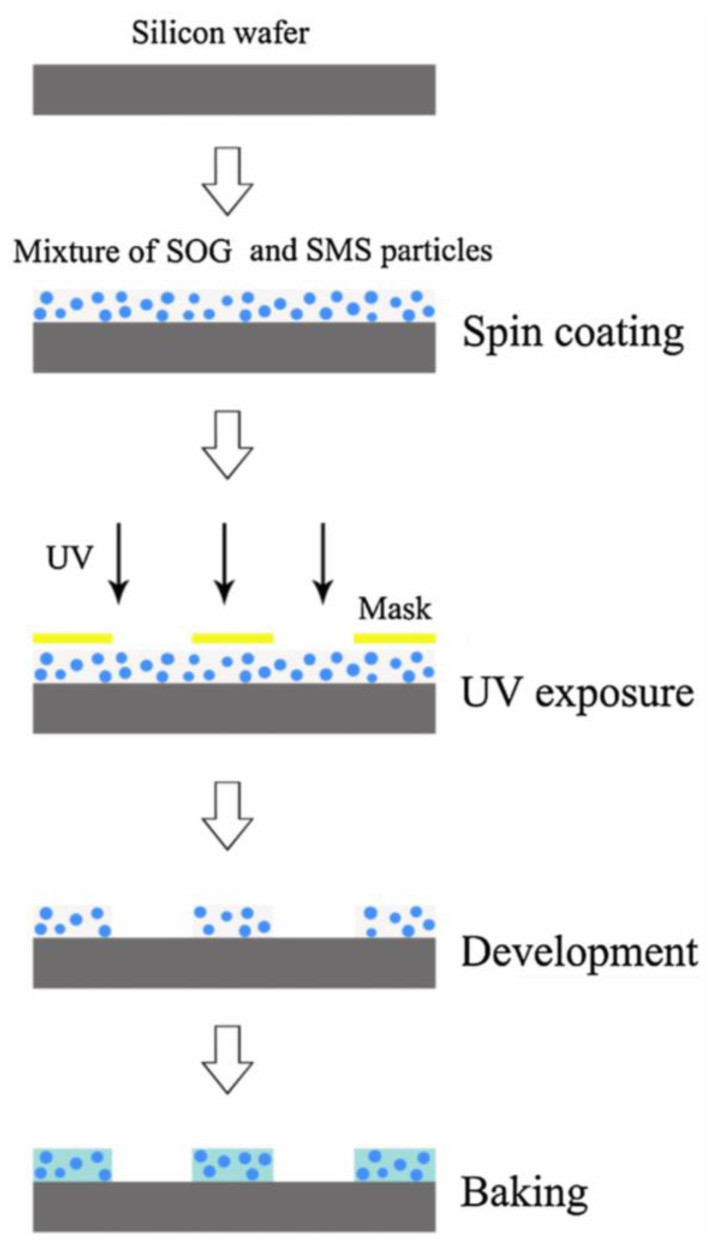
Schematic diagram of the experimental process used for SOG/SMS composite films. Adapted with permission from ref. [41]. Copyright 2016 Science Direct.

**Figure 7 polymers-14-00851-f007:**
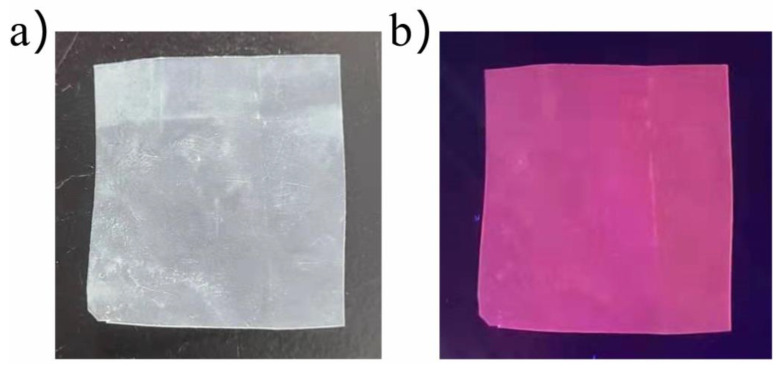
Photographs of the Moflight conversion material under (**a**) sunlight and (**b**) UV light. The foils measured approximately 2 × 2 cm^2^.

**Table 1 polymers-14-00851-t001:** The fluorescence quantum efficiency at solid state and light transmittance in films of TRZ, TRZME, and TRZOME.

Light Transmittance	TRZ	TRZEM	TRZOME	PVC
UV light area (200–380 nm)	0.45	0.28	0.30	0.69
Visible light area (380–800 nm)	0.84	0.90	0.89	0.89

**Table 2 polymers-14-00851-t002:** Mechanical properties of PLA/PBAT and PLA/PBAT/Eutt films. Adapted with permission from ref. [28]. Copyright 2018 Wiley Online Library.

PLA/PBAT/EuTT (wt/wt/wt)	Tensile Strength MD/TD (MPa)	Elongation at Break MD/TD (%)	Young’s Modulus MD/TD (MPa)	Tear Strength MD/TD (kN m^−1^)
35/65/0	33.4/25.8	442.5/422.6	685.8/307.1	181.0/184.2
35/65/0.3	36.7/25.2	462.8/483.0	766.0/272.6	157.1/173.2
35/65/0.3	33.3/20.1	535.8/413.6	795.2/256.6	148.3/162.4
35/65/0.5	33.1	484.3/396.1	1037.9/348.9	116.3/160.4

**Table 3 polymers-14-00851-t003:** Mechanical properties of PLA/PBAT, PLA/PBAT/EuDC, and PLA/PBAT/EuTT films. Adapted with permission from ref. [29]. Copyright 2019 Science Direct.

Sample	Elongation at Break MD/TD (%)
PLA/PBAT	442.5/422.6
PLA/PBAT/0.1%EuDC	473.8/448.0
PLA/PBAT/0.3%EuDC	557.8/451.4
PLA/PBAT/0.5%EuDC	595.0/460.9
PLA/PBAT/0.1%EuTT	462.8/483.0
PLA/PBAT/0.3%EuTT	535.8/413.6
PLA/PBAT/0.5%EuTT	484.3/396.1

## Data Availability

Not applicable.

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
