# Peer review of "Research Progress of Light Wavelength Conversion Materials and Their Applications in Functional Agricultural Films"

_polymers, 2022, doi:10.3390/polym14050851_

Round 1

Reviewer 1 Report

The paper introduces three common light conversion agents of fluorescent dyes, organic rare earth complexes and inorganic rare earth complexes based on the types of light conversion agents as well as their use in agricultural films and the mechanism of light conversion. At the same time, the preparation methods, advantages, disadvantages, and existing problems of various light conversion agents are classified and explained. The paper could be considered for publication after revision; however it is not very close to polymers field and would be better suited for some other journal ? The final decision about suitability for “Polymers” should be made by Editor ?

- “It's found that by using PE film coated with fluorescent dye 1, the lettuce had a significant increase in the fresh weight, dry weight, and leaf area after three weeks of culture.” Could the authors add some explanation for the described advantages ?

-Authors of the manuscript should prepare conclusions for the paper and describe the best light conversion agents, also advantages and disadvantages of all the agents of the different groups.

-The authors should explain why they think that the paper is suitable for “Polymers” journal ?

Author Response

Dear Reviewers:

Thank you for your letter and for the reviewers’ comments concerning our manuscript entitled “Research progress of light wavelength conversion materials and their applications in functional agricultural films” (polymers-1579394). Those comments are all valuable and very helpful for revising and improving our paper, as well as the important guiding significance to our research. We have studied comments carefully and have made correction which we hope meeting with approval. Revised portion are marked in the paper. The main corrections in the paper and the responds to the reviewer’s comments are as flowing:

Responds to the reviewer’s comments:

Comment from reviewer 1:

The paper introduces three common light conversion agents of fluorescent dyes, organic rare earth complexes and inorganic rare earth complexes based on the types of light conversion agents as well as their use in agricultural films and the mechanism of light conversion. At the same time, the preparation methods, advantages, disadvantages, and existing problems of various light conversion agents are classified and explained. The paper could be considered for publication after revision; however it is not very close to polymers field and would be better suited for some other journal ? The final decision about suitability for “Polymers” should be made by Editor ?

Comment 1

- “It's found that by using PE film coated with fluorescent dye 1, the lettuce had a significant increase in the fresh weight, dry weight, and leaf area after three weeks of culture.” Could the authors add some explanation for the described advantages?

Response:

       The explanation for the described advantages has been added in page 3.

The spectrum of sunlight arriving the earth's surface is between 290~3000 nm. Different components of light have different effects on plant growth. The blue violet light with the wavelength of 400~480 nm can be strongly absorbed by chlorophyll and carotenoids to promote the growth of plant stems and leaves; The red orange light with the wavelength of 600~700 nm is absorbed by chlorophyll, which can promote the fruit growth of plants; Yellow and green light with wavelength of 500~600 nm has little contribution to photosynthesis; The near ultraviolet light with the wavelength of 315~400 nm will make the plants shorter and the leaves thicker; Ultraviolet rays of 290~315 nm are harmful to most plants and can also lead to the aging of PE films. Else, in photosynthesis, plants absorb sunlight through chlorophyll, decompose water and release hydrogen (H+ and e -), reducing the absorbed carbon dioxide to organic matter. In this process, carbon dioxide and water are transformed into organic matter, and solar energy is transformed into chemical energy, which is stored in organic matter. And the light conversion films can accelerate this process in plants.

Comment 2

-Authors of the manuscript should prepare conclusions for the paper and describe the best light conversion agents, also advantages and disadvantages of all the agents of the different groups.

Response:

The conclusion part has been added in page 13.

 Conclusions

Rare earth inorganic complex light conversion agent has strong anti UV aging function, but its luminous efficiency is relatively weak. It is suitable for the preparation of long-life light conversion films and used for crops with low demand for light intensity; Organic rare earth complex light conversion agent has high light conversion efficiency, good matching with host polymers and relatively high preparation cost. It is an important direction for the future development of light conversion agent and suitable for the preparation of high-performance light conversion films; Organic conjugated small molecule light conversion agent has wider absorption spectrum, poor light conversion efficiency and aging resistance. It is more suitable for crops with special needs of spectrum.

Comment 3

-The authors should explain why they think that the paper is suitable for “Polymers” journal?

Response:

Light conversion agent is a very important polymer functional additive; The development of this additive can improve the properties of host polymer materials and expand their application fields; Although this review does not involve the synthesis, processing, and modification of polymers, it provides scientific and technological support for the high-quality application of 5 million tons of polyethylene materials for agricultural films every year.

Reviewer 2 Report

The review paper titled "Research progress of light wavelength conversion materials and their applications in functional agricultural films" is relatively well written. However, there are present some technical issues which have to be fixed before the paper acceptance:

  1. Fig. 3 should be provided with size bar or at least information about image size given in figure caption.
  2. Table 2 (mechanical properties). Mentioned values should be given with their errors by means of standard deviation. At least some statistical analysis has to be done.
  3. Fig. 7: size of the foils should be implemented into figure caption.

All in all, a minor revision of the manuscript is needed prior its acceptance.

Author Response

Dear Reviewers:

Thank you for your letter and for the reviewers’ comments concerning our manuscript entitled “Research progress of light wavelength conversion materials and their applications in functional agricultural films” (polymers-1579394). Those comments are all valuable and very helpful for revising and improving our paper, as well as the important guiding significance to our research. We have studied comments carefully and have made correction which we hope meeting with approval. Revised portion are marked in the paper. The main corrections in the paper and the responds to the reviewer’s comments are as flowing:

Responds to the reviewer’s comments:

Comment from reviewer 2:

The review paper titled "Research progress of light wavelength conversion materials and their applications in functional agricultural films" is relatively well written. However, there are present some technical issues which have to be fixed before the paper acceptance:

Comment 1

Fig. 3 should be provided with size bar or at least information about image size given in figure caption.

Response:

Figure 3 only shows the luminescence state of rare earth organic complex under different light environments. Its luminescence performance is only related to the light environment, but has little to do with the morphology and size of the complex. Therefore, the author only selected the photos of normal size and did not mark their size

Comment 2

Table 2 (mechanical properties). Mentioned values should be given with their errors by means of standard deviation. At least some statistical analysis has to be done.

Response:

These data are sorted out from the original literature, and the original literature does not give error and statistical analysis. As a review article, we can’t give the error and statistical analysis too.

Comment 3

Fig. 7: size of the foils should be implemented into figure caption.

Response:

Size of the foils has been implemented in the figure caption in figure 7.

Round 2

Reviewer 1 Report

If editor and other reviewers agree that the paper is suitable for “Polymers” I recommend the paper for publication after the revision.